# A systematic review on the prognostic role of radiologically-proven sarcopenia on the clinical outcomes of patients with acute pancreatitis

**Khang Duy Ricky Le**[1,2]*, **Harsh Patel**[1], **Emma Downie**[3]

1 Department of General Surgical Specialties, The Royal Melbourne Hospital, Melbourne, Victoria, Australia, 2 Geelong Clinical School, Deakin University, Geelong, Victoria, Australia, 3 Department of Hepatobiliary and Upper Gastrointestinal Surgery, St Vincent's Hospital Melbourne, Melbourne, Victoria, Australia

* khangduy.le@mh.org.au

## Abstract

### Background

Sarcopenia is a known risk factor for poor prognosis in chronic pancreatitis, however the impact of sarcopenia in acute pancreatitis (AP) is unknown. This systematic review examines the prognostic impact of sarcopenia on clinical outcomes in patients with acute pancreatitis.

### Methods

A systematic literature of Medline, EMBASE, Cochrane, and the World Health Organisation International Clinical Trials Registry Platform was undertaken to identify articles relating to sarcopenia, AP, and computed tomography imaging. Data collected was related to studies' demographic population, presence of sarcopenia, sarcopenia assessment methodology, obesity, pancreatitis severity, and short- and long-term complications of AP.

### Results

A total of four out of 114 unique peer-review articles were included in this review, encompassing 947 patients in total. Of the analysable data, 200 patients had sarcopenia and 640 did not. There was marked heterogeneity in the determination of the presence of sarcopenia between studies. No significant association was found between sarcopenia and pancreatic necrosis, organ failure, venous thromboembolism, recurrent acute pancreatitis, or mortality.

### Conclusion

Sarcopenia remains highly prevalent in patients suffering from acute pancreatitis. There is insufficient evidence to suggest sarcopenia is associated with poorer

**Data availability statement:** All relevant data are within the manuscript and its Supporting Information files.

**Funding:** The author(s) received no specific funding for this work.

**Competing interests:** The authors have declared that no competing interests exist.

outcomes in patients with acute pancreatitis. More high-powered studies are required to further characterise the impact of sarcopenia on patients with acute pancreatitis.

## Introduction

Acute pancreatitis (AP) is a significant inflammatory condition of the pancreas, commonly precipitated by gallstones, alcohol consumption, trauma and a myriad of other less common aetiologies [1]. The incidence of AP has been increasing since 2007, accounting for up to 20,000 acute hospital admissions per calendar year. This is hypothesised to be due to increased obesity, driving underlying determinants including cholelithiasis or hypertriglyceridaemia [2]. Despite increases in hospital admission rates, the mortality rate associated with AP is decreasing due to advances in medical technology, enabling more effective organ support for critically ill patients, as well as the transition to minimally invasive techniques for interventions such as necrosectomies [3,4]. Several scoring methods such as the Atlanta classification system exist to classify the severity of AP at the time of admission into mild, moderate and severe. However, these systems exhibit low specificity and poor prognostic accuracy in predicting AP-related complications [5,6].

Radiologically-assessed muscle mass has emerged as a surrogate marker for physical frailty, diminished functional status and as a predictor for poor post-operative outcomes [7,8]. Commonly, computed tomography (CT) parameters such as the psoas muscle area (PMA), psoas muscle index (PMI), total abdominal muscle area (TAMA), skeletal muscle attenuation (SMA), and skeletal muscle index (SMI) have been utilised to identify sarcopenia [9,10]. In most studies, these measurements are taken at the level of the third lumbar vertebrae so that psoas, paraspinal, and abdominal wall muscles are included in the same image, and changes at this level seem to correlate with whole-body changes in muscle mass [11]. Most importantly, sarcopenia identified in this way has been demonstrated to predict longer length of hospital stays, higher complication rates following pancreatectomy or islet cell autotransplantation and increased overall mortality in patients with chronic pancreatitis [12,13]. Given CT is frequently used in the diagnosis of AP and for assessment of potential complications, there is an opportunity for clinicians to also leverage these scans to identify sarcopenia [6]. Despite this, the prognostic impact of radiologic sarcopenia in patients with AP remain poorly characterised. This systematic review evaluates the prognostic impact of radiologically-proven sarcopeniaon clinical outcomes in patients admitted with AP.

## Methods

### Search protocol and registration

This systematic review was performed in adherence to the Preferred Reporting Items for Systematic Review and Meta-Analyses (PRISMA) guidelines (S1 Checklist) [14]. The review protocol was prospectively registered in the PROSPERO database (PROSPERO ID: CRD42024566823).

## Literature search

A literature search was conducted on Medline, EMBASE, Cochrane CENTRAL and the World Health Organisation International Clinical Trials Registry Platform (ICTRP) on 9 July 2024. Additional relevant articles were identified by hand-searching the reference lists of captured articles. The search strategy combined medical subject headings (MeSH) and keywords related to sarcopenia, acute pancreatitis and computed tomography imaging. Boolean operators and truncations were utilised in the search to identify all relevant articles. The full search strategy can be found in the S1 Appendix.

## Eligibility criteria

Peer-reviewed full-text articles available in the English language and which evaluated the influence of sarcopenia on outcomes of patients with AP were considered. Articles were included if they were: 1. Original, peer-reviewed randomised-controlled trials, prospective or retrospective non-randomised observational studies; 2. Articles that included adults (aged 18 years or older) diagnosed with AP and had evaluation for sarcopenia with cross-sectional imaging in the same admission; 3. Articles that utilised high modality cross-sectional imaging (such as CT or MRI) area or density-based measurement methods including skeletal muscle index (SMI), psoas muscle index (PMI), total psoas index (TPI), total psoas area (TPA), total abdominal muscle area or skeletal muscle density (SMD); 4. Articles that evaluated the outcomes of interest. Articles were excluded if they were: 1. Of the following study designs and types; systematic reviews, meta-analyses, non-human studies, preclinical studies, conference papers, letters, editorials, opinion articles, comments, case reports or case series; 2. Articles that evaluated paediatric populations (age less than 18 years); 3. Articles with incomplete data; 4. Articles that evaluated sarcopenia by other methods such as skinfold thickness testing, arm or torso circumference testing, total body potassium levels or daily urinary creatinine output; 5. Articles that evaluated other types of pancreatitis including chronic pancreatitis, and 6. Articles that did not explore the outcomes of interest.

## Literature screening

Initial screening by title and abstract was performed by two independent investigators (KDRL, HP). Titles and abstracts that had insufficient information proceeded to full-text analysis. The same two investigators then independently performed full-text analysis of articles based on eligibility criteria for inclusion. Disagreement during this process was resolved by discussion and consensus. A table of all studies that underwent full-text analysis is available in the supplementary data (S1 Table).

## Outcomes

Endpoints of interest assessed outcomes following AP including length of stay, intensive care unit (ICU) admission, ICU length of stay, pseudocyst, necrosis, respiratory failure, acute kidney injury (AKI), organ failure, liver failure, need for dialysis or filtration, deep vein thrombosis (DVT), superior mesenteric vein (SMV) thrombosis, portal vein (PV) thrombosis, venous thromboembolic (VTE) phenomenon (a composite for DVT, SMV thrombosis, splenic vein thrombosis and PV thrombosis), total complications, severe complications (Clavien Dindo 3+), recurrence as well as survival outcomes including overall mortality and in-hospital mortality.

## Data extraction

Included articles were extracted for data including study name, author, study design, year and country of publication, demographic data, presence or absence of sarcopenia, obesity, severity of pancreatitis as well as data relevant to the outcomes of interest. Data extraction parameters are presented in the supplementary data (S1–S6 Tables).

### Risk of bias assessment

Articles were assessed for methodological quality using the Risk Of Bias In Non-randomised Studies - of Interventions (ROBINS-I) tool by two independent investigators (KDRL, HP) [15]. Disagreement during this process was resolved by discussion and consensus. Publication bias was assessed with a funnel plot as part of the statistical analysis of included articles if sufficient numbers of publications (>25) were available).

### Statistical analysis

Statistical analysis was performed utilising Review Manager 5.4 (RevMan 5.4) (Cochrane, London, United Kingdom). Odds ratios (OR), risk ratio (RR), hazard ratios (HR) and their 95% confidence intervals (95% CI) were extracted from the included studies and meta-analysed if there was homogeneous data where possible. In the absence of homogeneous data, descriptive results were presented. Continuous data whenever relevant was converted to single measures of effect, such as from median and interquartile range to mean and standard error utilising the Wan method [16].

Heterogeneity between studies was evaluated with the Higgins $I^2$ test [17]. Values of $I^2$ at 25%, 50%, and 75% were graded as low, moderate and high heterogeneity respectively. $Tau^2$ was calculated using the Restricted Maximum-Likelihood method. A fixed-effects model was used if substantial heterogeneity was absent and a random-effects model if substantial heterogeneity was found. A p-value of $< 0.05$ was considered to be statistically significant. To determine sources of heterogeneity, subgroup analyses and meta-regression analyses whenever relevant were performed.

### Subgroup analyses

To investigate confounding factors, clinically relevant subgroup analyses planned include evaluation of outcomes stratified by risk of bias of included articles, presence or absence of obesity in sarcopenic patients, aetiology of pancreatitis (gall-stone, alcohol and other) and severity of pancreatitis.

### Assessment of certainty

Results from statistical analysis were evaluated for certainty of evidence by two independent investigators (KDRL, HP) using the transparent Grading of Recommendations, Assessment, Development and Evaluations (GRADE) framework. Disagreement during this process was resolved by consensus.

## Results

### Literature search

The literature search work flow is demonstrated below (Fig 1). 114 unique peer-reviewed articles were identified from the literature search. 105 articles were excluded following screening by title and abstract. Of the nine remaining studies that underwent full-text analysis according to eligibility criteria, four articles were eligible for inclusion in this review.

### Overview of included studies and patient demographics

An overview of the included studies and patient characteristics are presented in Table 1. Four studies encompassing 947 patients were included [18–21]. Notably, from these cohorts, data was analysable for 839 patients. Overall, the pooled mean age was 49.73 years (standard deviation 15.24) with 622 males and 325 females. The most common aetiology of pancreatitis was biliary pathology (n = 279). Obesity status was poorly reported. Of the analysable patient data, there were 200 patients with sarcopenia and 640 patients without sarcopenia.

### Methods of sarcopenia determination

Sarcopenia was determined by cross-sectional imaging with CT in all included studies (Table 2). The predominant method reported by authors was using CT at the level of the third lumbar vertebra (L3). There was marked

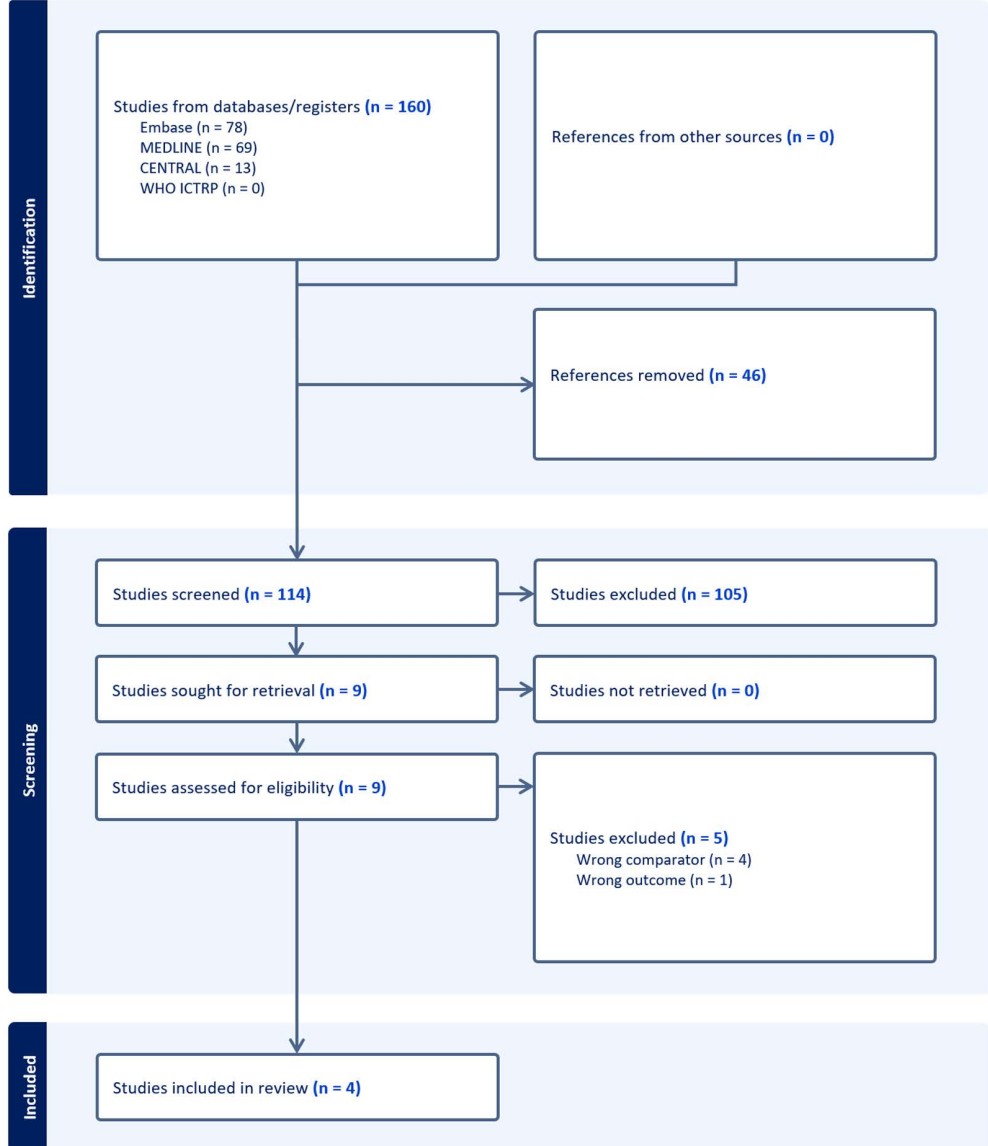

**Fig 1. Workflow of literature search and article selection.**

heterogeneity in the reference method to determine the presence of sarcopenia, with the most common index being psoas muscle index [22–26].

## Sarcopenia status and outcomes following admission for acute pancreatitis

**Pancreatic necrosis.** Two studies analysed the presence of necrosis [18,20]. A random-effects model was employed to compare sarcopenic to non-sarcopenic patients. The pooled analysis of 374 patients (118 sarcopenic, 256 non-sarcopenic) demonstrated that there was little to no evidence to suggest sarcopenia is a prognostic factor for necrosis (RR 0.92, 95% CI 0.71–1.20, p = 0.55, $I^2$ = 65%) (Fig 2).

**Table 1. Overview of included studies and patient demographics.**

| Study | Study design | Country of Publication | Sample size | Age | Sex | Sarcopenia Status | Severity of Pancreatitis | Aetiology of Pancreatitis | Obesity |
|---|---|---|---|---|---|---|---|---|---|
| *Author, Year* | | | *N* | *Mean +/- SD* | *M, F* | *S, NS* | | | *N* |
| Farquhar, 2023 [18] | Prospective cohort study | United Kingdom | 141 (74 with analysable data) | 59 +/-16.63 | 85, 56 | S: 44, NS: 30 | NR | Biliary: 55 (S: 41, NS: 14) Alcohol: 40 (S: 31, NS: 9) Other: 46 | 77 (S: 67, NS: 11) |
| Fu, 2023 [19] | Retrospective cohort study | China | 269 | 50 +/-20.12 | 158, 111 | S: 29, NS: 241 | Mild: 190 Moderate: 55 Severe: 24 | Biliary: 69 Alcohol: 78 HTG: 47 Other: 75 | NR |
| Yee, 2021 [20] | Retrospective cohort study | United States of America | 341 (300 with analysable data) | 51 +/- 14 | 221, 120 | S: 74, NS: 226 | Mild: 40 Moderate: 86 Severe: 215 | Biliary: 155, Alcohol: 87, HTG: 26, Other: 73 | NR |
| Yu, 2021 [21] | Retrospective cohort study | China | 196 | 40.52 +/-9.72 | 158, 38 | S: 53, NS: 143 | NR | HTG: 196 | 20 |

Abbreviations: S, sarcopenia; NS, non-sarcopenia; NR, not reported; HTG, hypertriglyceridaemia.

**Table 2. Overview of methods of sarcopenia determination and reference methods utilised by included studies.**

| Study *Author, Year* | Modality of Assessment | Index | Cut-off Values | Reference method |
|---|---|---|---|---|
| Farquhar, 2023 [18] | CT L3 | SMI, PMI and SMA | SMI: M: <43 cm2/m2 (for BMI <25) or <53 cm2/m2 (for BMI >25) F: <41 cm2/m2 PMI: M: <5.9 cm2/m2 F: <4.1 cm2/m2 SMA: M: <33.9 HU F: <30.9 HU | Martin (22), Okumura (24), Van Dijk (23) |
| Fu, 2023 [19] | CT L3 | PMI | M: <3.85 cm2/m2 F: <3.20 cm2/m2 | In house |
| Yee, 2021 [20] | CT L4 | PMI | M: <3.46 cm2/m2 F: <2.86 cm2/m2 | Yoo (25) |
| Yu, 2021 [21] | CT L3 | SMA | M: <52.4 cm2/m2 F: <38.5 cm2/m2 | Prado (26) |

Abbreviations: CT, computed tomography; L3, third lumbar vertebra; L4, fourth lumbar vertebra; SMI, skeletal muscle index; PMI, psoas muscle index; SMA, skeletal muscle area.

**Organ failure.** Two studies analysed the presence of organ failure [19,20]. A random-effects model was employed to compare sarcopenic to non-sarcopenic patients. The pooled analysis of 570 patients (103 sarcopenic, 467 non-sarcopenic) demonstrated that there was little to no evidence to suggest sarcopenia is a prognostic factor for organ failure (RR 1.24, 95% CI 0.51–2.99, p = 0.63, $I^2$ = 84%) (Fig 3).

**Venous thromboembolic phenomena.** Two studies analysed the presence of VTE phenomena [18,19]. A random-effects model was employed to compare sarcopenic to non-sarcopenic patients. The pooled analysis of 344 patients (73 sarcopenic, 271 non-sarcopenic) demonstrated that there was little to no evidence to suggest sarcopenia is a prognostic factor for VTE phenomena (RR 1.88, 95% CI 0.22–16.35, p = 0.57, $I^2$ = 86%) (Fig 4).

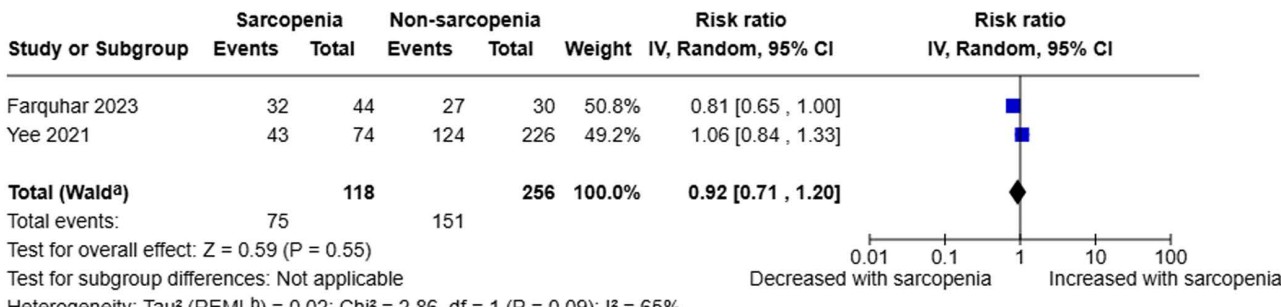

**Fig 2. Forest plot evaluating presence of pancreatic necrosis in sarcopenic compared to non-sarcopenic patients with acute pancreatitis utilising a random effects model.**

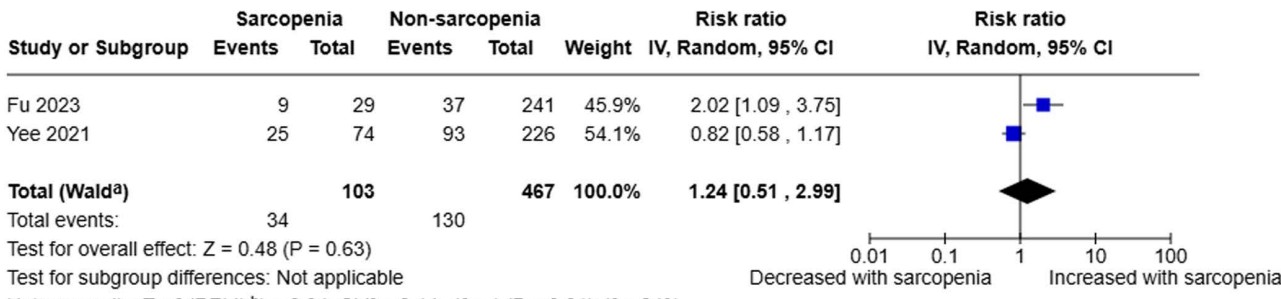

**Fig 3. Forest plot evaluating presence of organ failure in sarcopenic compared to non-sarcopenic patients with acute pancreatitis utilising a random effects model.**

**Recurrent acute pancreatitis.** Two studies analysed the presence of recurrent AP [18,21]. A random-effects model was employed to compare sarcopenic to non-sarcopenic patients. The pooled analysis of 270 patients (97 sarcopenic, 173 non-sarcopenic) demonstrated that there was little to no evidence to suggest sarcopenia is a prognostic factor for recurrent AP (RR 0.98, 95% CI 0.67–1.43, p=0.90, I²=0%) (Fig 5).

**Mortality.** Two studies analysed mortality [18,20]. A random-effects model was employed to compare sarcopenic to non-sarcopenic patients. The pooled analysis of 374 patients (118 sarcopenic, 256 non-sarcopenic) demonstrated that there was little to no evidence to suggest sarcopenia is a prognostic factor for mortality (RR 0.84, 95% CI 0.15–4.71, p=0.84, I²=76%) (Fig 6).

**Other outcomes.** Pooled analysis was not possible for other outcomes of interest as identified *a priori*. Despite this, individual studies reported on relevant outcomes including length of stay, intensive care unit (ICU) admission, ICU length of stay and PV thrombosis. Farquhar *et al.* demonstrated that there was little to no evidence to suggest length of stay differed between sarcopenia and non-sarcopenic cohorts (mean difference (MD) -16.00, 95% CI -95.48–63.48,

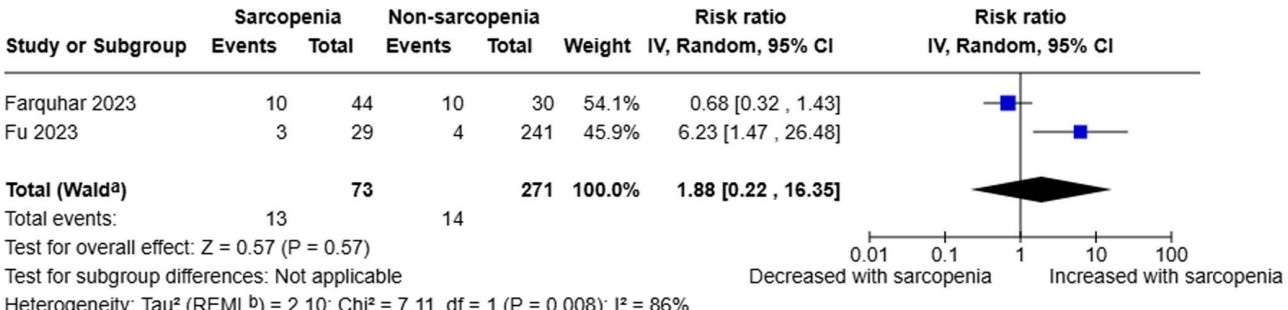

Fig 4. Forest plot evaluating presence of venous thromboembolic phenomena in sarcopenic compared to non-sarcopenic patients with acute pancreatitis utilising a random effects model.

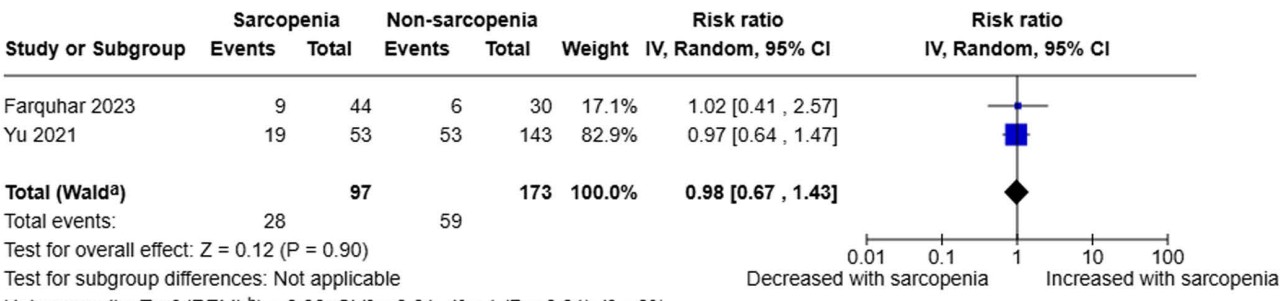

Fig 5. Forest plot evaluating presence of recurrent acute pancreatitis in sarcopenic compared to non-sarcopenic patients with acute pancreatitis utilising a random effects model.

p = 0.69) [18]. Fu *et al.* demonstrated that there was little to no evidence to suggest ICU admission rates differed between sarcopenia and non-sarcopenic cohorts (OR 0.92, 95% CI 0.11–7.54, p = 0.94) [19]. When examining length of stay duration in the ICU, Farquhar *et al.* demonstrated that there was little to no evidence to suggest this differed between sarcopenia and non-sarcopenic cohorts (MD -14.66, 95% CI -49.19–19.87, p = 0.41) (18). Lastly, Farquhar et al. demonstrated that there was little to no evidence to suggest rates of PV thrombosis differed between sarcopenia and non-sarcopenic cohorts (OR 0.59, 95% CI 0.21–1.66, p = 0.32) [18].

There was no data available for other *a priori* outcomes including pseudocyst, respiratory failure, AKI, liver failure, need for dialysis or filtration, DVT, SMV thrombosis, total complications, severe complication and in-hospital mortality.

## Risk of bias assessment

Overall, 3 articles (75%) were considered to have moderate risk of bias and 1 article was considered to be at serious risk of bias (25%) (Fig 7). Key areas that introduced bias include the presence of selection bias due to including only

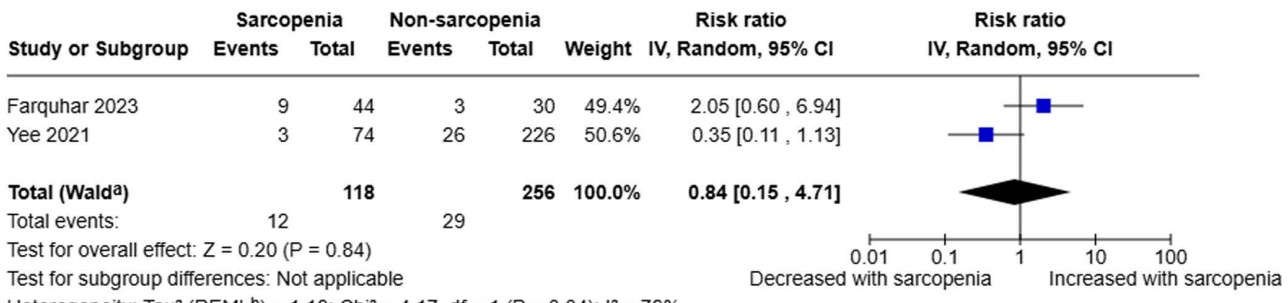

Fig 6. Forest plot evaluating mortality in sarcopenic compared to non-sarcopenic patients with acute pancreatitis utilising a random effects model.

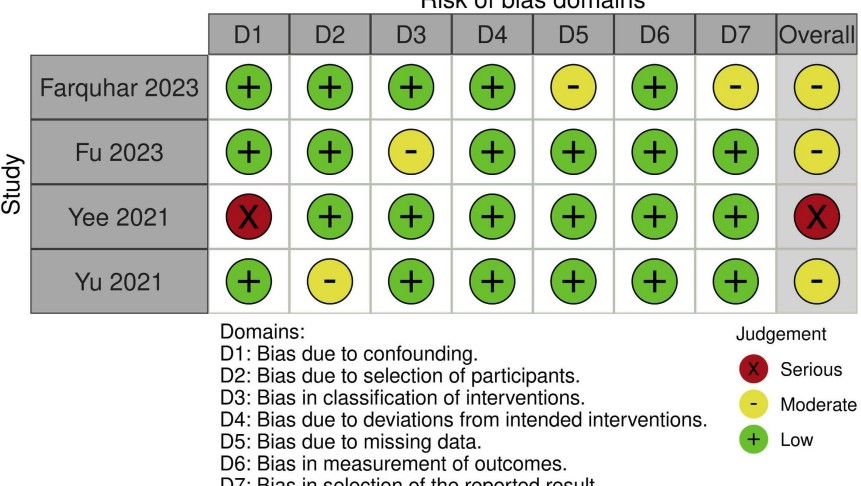

Fig 7. Risk of bias assessment of included articles utilising the ROBINS-I tool.

hypertriglyceridaemic AP cases or cases of necrotising pancreatitis [20,21]. Furthermore, there was additional bias introduced due to proprietary cut-offs for sarcopenia in one article [19]. A funnel plot was not performed given the low number of included articles.

## Discussion

This systematic review and meta-analysis demonstrated insufficient evidence to suggest sarcopenic patients with AP are at greater risk of complications including pancreatic necrosis, organ failure, venous thromboembolic phenomenon, recurrent AP or death. Furthermore, we demonstrate that there is a significant paucity of evidence for other clinically relevant outcomes related to hospital admission for episodes of AP. Additionally, these findings are made with very low to low certainty as per our GRADE assessment (Table 3). This was attributable to clinical heterogeneity of results, risk of bias and in some cases, imprecision of results. To evaluate the effect of confounding variables in these non-committal results,

**Table 3. GRADE assessment of outcomes.**

| Outcomes | Effect size (RR and 95% CI) | Certainty of evidence (GRADE) | Comments |
|---|---|---|---|
| Pancreatis necrosis | 0.92 [0.71, 1.20] (Two studies) | ⊕◯◯◯ Very low | Downgrade 1 point for risk of bias, downgrade 1 point for inconsistency due to clinical heterogeneity, downgrade 1 point for imprecision. Publication bias unable to be ascertained due to paucity of data |
| Organ failure | 1.24 [0.51, 2.99] (Two studies) | ⊕◯◯◯ Very low | Downgrade 1 point for risk of bias, downgrade 1 point for inconsistency due to clinical heterogeneity, downgrade 1 point for imprecision. Publication bias unable to be ascertained due to paucity of data |
| Venous thromboembolic phenomena | 1.88 [0.22, 16.35] (Two studies) | ⊕◯◯◯ Very low | Downgrade 1 point for risk of bias, downgrade 1 point for inconsistency due to clinical heterogeneity, downgrade 1 point for imprecision. Publication bias unable to be ascertained due to paucity of data |
| Recurrent acute pancreatitis | 0.98 [0.67, 1.43] (Two studies) | ⊕◯◯◯ Low | Downgrade 1 point for risk of bias, downgrade 1 point for inconsistency due to clinical heterogeneity. Publication bias unable to be ascertained due to paucity of data |
| Mortality | 0.84 [0.15, 4.71] (two studies) | ⊕◯◯◯ Very low | Downgrade 1 point for risk of bias, downgrade 1 point for inconsistency due to clinical heterogeneity, downgrade 1 point for imprecision. Publication bias unable to be ascertained due to paucity of data |

planned subgroup analyses such as analysis after removal of articles at high risk of bias, evaluation of the effect of sarcopenic obesity as well as sensitivity analysis based on aetiology and severity of pancreatitis was planned. Despite this, there was insufficient evidence for these analyses to be performed.

Sarcopenia has been demonstrated to be highly prevalent in populations with severe AP and also implicated as a predictor of poorer outcome in those with chronic pancreatitis [13,18]. Given this, we hypothesised that sarcopenia may be a prognostic factor leading to increased adverse outcomes for patients with AP. Our study however was unable to demonstrate this effect. One explanation for this is related to the quality of the underlying studies, which are limited by small sample sizes and therefore may be underpowered to derive a meaningful effect. Another explanation may be related to the severity of pancreatitis and this relationship with sarcopenia. For example, on examining individual studies, Farquhar *et al.* demonstrated that in individuals with severe AP, sarcopenia and sarcopenic obesity was associated with poorer outcomes including increased mortality and organ injury rates [18]. Notably, our study included patients of predominantly mild to moderate severity where reported, with half the studies not reporting severity. Sensitivity analysis by stratifying based on severity therefore was not possible in our review due to paucity of data and thus the interplay between sarcopenia and AP outcomes remains unexplored.

When evaluating the literature on chronic pancreatitis populations, there is evidence to suggest that CT-proven sarcopenia is associated with greater rates of complication and poorer survival [12]. Specifically, Bundred *et al.* in a systematic review of sarcopenia as a prognostic factor for patients with chronic pancreatitis, identified 9 cohort studies reporting on 977 patients [12]. In this review, the authors demonstrated sarcopenia was associated with greater mortality within 1 year (HR 6.69, 95% CI 1.79–24.9, p < 0.001) [12]. Of these studies, Olesen *et al.* was noted to be the only cohort study with prospective data [13]. In this study of 182 chronic pancreatitis patients, they demonstrated sarcopenia was a prognostic indicator for increased length of hospital stay (23.4 +/- 36.4 days vs 8.1 +/-15.9 days, p < 0.001) and reduced survival (HR 6.7, 95% CI 1.8–25.0, p = 0.005) [13]. It is postulated that this may be attributed to the underlying pathogenesis of sarcopenia for this population. For example, it is hypothesised that malnutrition, increased levels of leptin due to repeated pancreas injury and ongoing low-grade inflammation may be factors that contribute to sarcopenia [27–31]. These factors also exist in patients with cancer and may also in part explain the presence of sarcopenia, as well as the poorer outcomes with sarcopenia status. It still remains unclear at which point sarcopenia becomes relevant to patient outcomes on the spectrum of acute to chronic pancreatitis. Nonetheless, nutritional interventions and addressing in-patient nutrition as well as

chronic malnutrition has been shown to improve outcomes for patients with both acute and chronic pancreatitis [32–34]. Nutritional supplementation remains an area of ongoing research in sarcopenia, however may play a role in preventing ongoing muscle wasting and potentially improve muscle mass [31]. Best-practice nutritional interventions for the indication of sarcopenia with concurrent acute pancreatitis remains underexplored. Future research should consider exploring this temporal relationship with longer periods of follow-up and the role of nutrition and its impact on this relationship, particularly for patients who have recurrent episodes of AP.

There has been notable recognition regarding the impacts of frailty and sarcopenia on patient outcomes. Understandably, this occurs in the context of ageing populations and the increasing desire to improve patient outcomes in light of this. There are however limitations to the accuracy and sensitivity of frailty assessment, with scoring systems such the Short Physical Performance Battery (SPPB) or evaluation of the Fried Phenotype relying on either subjective reporting, or individual functional performance at the time of investigation [35–37]. Sarcopenia has been demonstrated to be objectively measured with cross-sectional imaging, with the presence of sarcopenia associated with clinically relevant outcomes. Specifically, sarcopenia identified on CT has been demonstrated to predict poorer outcomes following surgery, including increased mortality rates and post-operative complications [38,39]. Poorer outcomes with sarcopenia are also identified outside of surgery, with reduced survival in patients undergoing radiotherapy and geriatric blunt trauma populations [40,41]. Interestingly these prognostic features are not demonstrated in all situations, with outcomes being similar irrespective of sarcopenia status on liver transplant recipients and individuals who undergo surgery for non-small cell lung cancer [42,43]. Overall, for patients with AP, despite the increased prevalence demonstrated in the literature, the relationship of AP and sarcopenia cannot be confidently established due to the limitations of the underlying evidence.

To the authors' knowledge, this is the first systematic review and meta-analysis to evaluate radiologically-proven sarcopenia and outcomes related to AP. However, there are limitations to note. Firstly, the majority of studies were retrospective observational studies (n = 3, 75%) with small sample size, therefore lacking key components of rigorous study design including randomisation, allocation concealment, cohort matching and blinding. In addition, the foundation of studies were deemed to be at moderate to serious risk of bias. Secondly, the included studies were highly heterogeneous due to varying sample sizes, demographic populations and inclusion criteria. The latter is an important limitation to recognise, with one study including only patients with necrotising pancreatitis and another including only patients with hypertriglyceridaemic AP, both potentially confounding factors for outcomes. When further considering the included population, there was a greater proportion of study participants with mild or moderate AP with the available reported data. These patients often experience improved outcomes when compared to those with severe AP in addition to a lower degree of sarcopenia. We attempted to control for effects of disease severity through planned subgroup analyses however this was not possible due to the paucity of data and therefore the impact of sarcopenia on separate entities of disease severity remain underexplored. Moreover, there was significant heterogeneity in the methods by which sarcopenia was determined, with varying muscle indices, cut-off values and reference methods employed. This finding is likely to be an additional confounding factor implicating the precision of results. Additionally, in all of the studies, cross-sectional imaging was performed either at point of index diagnosis of AP or after a period of hospital admission which was usually a week. Only one study protocolised serial radiological assessment of sarcopenia during admission which demonstrated that sarcopenia as identified by PMI progressed by 14% over 30 days in the necrotising pancreatitis cohort [20]. This change identifies another clinically relevant research question about the dynamic nature of body composition changes and sarcopenia during episodes of acute pancreatitis and their impacts on patient outcomes. Despite this, more mild-moderate cases of acute pancreatitis, such as those of the included studies within this review, are unlikely to experience this degree of muscle wasting, nor experience a length of stay long enough for serial testing to be performed. Therefore, this raises two considerations with our review. Firstly, although our systematic review explores the prognostic impact of sarcopenia identified close to admission on clinical outcomes during hospitalisation, the effect of dynamic changes in body composition based on severity of acute pancreatitis on clinical outcomes remains a poorly characterised area. Secondly, given many patients from

the included studies had cross-sectional imaging not at the precise moment of admission, there is the potential for the confounding effect of accelerated sarcopenia development due to hospitalisation and the underlying acute pancreatitis. Despite this, although many of the included cases were imaged within the week and were not cases of severe or necrotising pancreatitis, the effect of the confounding variable should be considered. Furthermore, patients with AP that suffer from complications often have comorbidities and multiple comorbidities may be a factor that drives sarcopenia development. Desptie this, the presence of comorbidities was poorly reported in the literature and may act as additional confounding factors that are not accounted. Lastly, given this review only included articles available in the English language, there is the potential for missing data that would have otherwise been available for evaluation from culturally and linguistically diverse settings. Therefore, the results of this meta-analysis should be considered with caution.

## Conclusion

This systematic review and meta-analysis demonstrates insufficient evidence to suggest CT-proven sarcopenia is with a prognostic marker of poorer clinical outcomes in patients with AP. This finding occurs in the presence of poor-quality evidence, with small sample sizes and marked heterogeneity. There is a need for further appropriately powered prospective trials to further characterise the impact of sarcopenia in those with AP.

## Supporting information

**S1 Checklist.  PRISMA Checklist.**
(DOCX)

**S1 Appendix.  Appendix A: Search strategy for databases and trial registries.**
(DOCX)

**S1 Table.  Table of included and excluded studies.**
(DOCX)

**S2 Table.  Data for organ failure of included studies.**
(DOCX)

**S3 Table.  Data for necrosis of included studies.**
(DOCX)

**S4 Table.  Data for recurrence of included studies.**
(DOCX)

**S5 Table.  Data for venous thromboembolic phenomenon of included studies.**
(DOCX)

**S6 Table.  Data for mortality of included studies.**
(DOCX)

## Acknowledgments

None

## Author contributions

**Conceptualization:** Khang Duy Ricky Le.

**Data curation:** Khang Duy Ricky Le, Harsh Patel.

**Formal analysis:** Khang Duy Ricky Le, Harsh Patel, Emma Downie.

**Investigation:** Khang Duy Ricky Le, Harsh Patel, Emma Downie.

**Methodology:** Khang Duy Ricky Le, Harsh Patel, Emma Downie.

**Project administration:** Khang Duy Ricky Le.

**Resources:** Khang Duy Ricky Le, Emma Downie.

**Software:** Khang Duy Ricky Le.

**Supervision:** Khang Duy Ricky Le, Emma Downie.

**Validation:** Khang Duy Ricky Le, Harsh Patel, Emma Downie.

**Visualization:** Khang Duy Ricky Le, Harsh Patel.

**Writing – original draft:** Khang Duy Ricky Le, Harsh Patel, Emma Downie.

**Writing – review & editing:** Khang Duy Ricky Le, Harsh Patel, Emma Downie.

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
