## [Decision Letter · Decision Letter 0]

29 Jan 2025

PONE-D-24-32431A systematic review on clinical outcomes associated with radiologically-proven sarcopenia on patients with acute pancreatitis.PLOS ONE

Dear Dr. Le,

Thank you for submitting your manuscript to PLOS ONE. After careful consideration, we feel that it has merit but does not fully meet PLOS ONE’s publication criteria as it currently stands. Therefore, we invite you to submit a revised version of the manuscript that addresses the points raised during the review process. **I have explained why we have only one review of this manuscript below.  Please primarily focus on the points I have made below that need to be corrected when you resubmit.  You should also reflect on the comments by Reviewer 1 and expand the Discussion appropriately. **

We look forward to receiving your revised manuscript.

Kind regards,

James M Wright

Academic Editor

PLOS ONE

**Journal Requirements:**

2. Please amend either the abstract on the online submission form (via Edit Submission) or the abstract in the manuscript so that they are identical.

4. As required by our policy on Data Availability, please ensure your manuscript or supplementary information includes the following: 

**Additional Editor Comments:**

We have had a very difficult time finding reviewers for this prognosis review of the effect of sarcopenia on adverse outcomes in patients with acute pancreatitis. Only 1 person out of over 40 people invited has provided a review. I think the difficulty is due to the fact that the question being asked is of little importance and interest. Despite that the review has some merit but will require significant improvements listed below.

1. This is a prognosis review and that must be reflected in the title and throughout. The authors appear not to appreciate this and refer to randomization etc. which are relevant to intervention reviews but are impossible for a prognosis review.

2. The authors imply in the background and throughout that the question of the effect of sarcopenia on mortality and other outcomes in chronic pancreatitis is answered with robust evidence. I do not accept that conclusion. The effect on mortality is based on one small short term trial. The actual evidence in chronic pancreatitis must be presented in detail so that readers can come to their own conclusions. As it is presented it is misleading and inaccurate.

3. The calculations of risk in Figures 2-6 are not done properly. You are attempting to estimate the degree to which sarcopenia increases the adverse outcomes compared to the absence of sarcopenia. The labels should reflect that such as increased with sarcopenia on the right and decreased with sarcopenia on the left. The number of events is high so using Risk ratio is more appropriate than odds ratio. Therefore numbers greater than 1 reflect poor prognosis and numbers less than one reflect the opposite.

4. The Figures 2-6 must include the outcome being measured.

5.  Table 3. Including the Effect size without 95% CI is meaningless for a reader. It should be RR with 95% CI demonstrating that none are close to statistically significant.

Reviewers' comments:

Reviewer's Responses to Questions

**Comments to the Author**

1. Is the manuscript technically sound, and do the data support the conclusions?

Reviewer #1: Yes

2. Has the statistical analysis been performed appropriately and rigorously? 

Reviewer #1: Yes

3. Have the authors made all data underlying the findings in their manuscript fully available?

Reviewer #1: Yes

4. Is the manuscript presented in an intelligible fashion and written in standard English?

Reviewer #1: Yes

5. Review Comments to the Author

**Reviewer #1:**  Dear Authors,

this systematic review is very well performed, methodologically sound and clearly written. The issue of sarcopenia in AP is very interesting, since sarcopenia has been associated with poorer patients outcomes in various diseases and conditions.

In the introduction part, I'd emphasize that Atlanta classification is not used to predict the severity of AP, but it's a classification system with clearly defined criteria for mild, moderate, and severe AP.

There is very little data on the impact of sarcopenia on AP outcomes, and it is not clear whether sarcopenia is a cause or a consequence in patients with severe pancreatitis. The fact is that more than 25% of patients with AP have sarcopenia on admission. However, it would be very important to determine to what extent sarcopenia changes during hospitalisation and how the possible worsening of sarcopenia affects the outcome in AP.

CT is a fairly reliable method for detecting sarcopenia, but it is very important to point out that CT in AP is very often not performed on admission of the patient, but only after a few days. The aforementioned period in severe AP can only worsen the sarcopenia compared to the period of the patient's admission to hospital or the onset of symptoms.

There is also the question of the appropriateness of nutritional intervention in these patients.

Most patients with AP, especially in retrospective analyses, actually have a mild to moderate form of the disease, and even when the authors state that the predicted severe forms of the disease are included in the study, a large proportion of these patients do not ultimately have severe AP. In these patients, sarcopenia is unlikely to have a significant impact on the outcome of the disease.

It would also be good to include the presence of comorbidities in the data analysis, because other diseases, especially chronic diseases, can significantly influence the presence of sarcopenia and the outcome of AP.

In addition to the limitations of your research already mentioned, I think the discussion should be expanded to include the above comments.

6. PLOS authors have the option to publish the peer review history of their article (what does this mean? ). If published, this will include your full peer review and any attached files.

**Do you want your identity to be public for this peer review?** For information about this choice, including consent withdrawal, please see our Privacy Policy .

Reviewer #1: No

---

## [Author Response · Author response to Decision Letter 1]

13 Mar 2025

We thank the editor and reviewer for their time an expertise in improving our manuscript. We have judiciously attended to the comments provided and are pleased to submit a revised manuscript in line with the goals and expectations of PLOS One. Please find attached to this submission a revised manuscript with tracked changes in addition to revised figures 2-6 and additional supplementary tables (1 and 2). Below are our responses to the editor and reviewer comments which are provided verbatim for ease of orientation.

Editor comments.

"This is a prognosis review and that must be reflected in the title and throughout. The authors appear not to appreciate this and refer to randomization etc. which are relevant to intervention reviews but are impossible for a prognosis review.:

We thank the editor for their comment. Indeed this is a systematic review on the prognostic impact of radiologically-proven sarcopenia in acute pancreatitis outcomes. We have amended our manuscript to account for this as suggested.

"The authors imply in the background and throughout that the question of the effect of sarcopenia on mortality and other outcomes in chronic pancreatitis is answered with robust evidence. I do not accept that conclusion. The effect on mortality is based on one small short term trial. The actual evidence in chronic pancreatitis must be presented in detail so that readers can come to their own conclusions. As it is presented it is misleading and inaccurate."

We thank the editor for their comments. We agree with this comment and have attended to the section of the discussion in question by improving the wording and supporting the statements made with the underlying evidence to provide the readership a clearer understanding of the underlying evidence in chronic pancreatitis patients.

"The calculations of risk in Figures 2-6 are not done properly. You are attempting to estimate the degree to which sarcopenia increases the adverse outcomes compared to the absence of sarcopenia. The labels should reflect that such as increased with sarcopenia on the right and decreased with sarcopenia on the left. The number of events is high so using Risk ratio is more appropriate than odds ratio. Therefore, numbers greater than 1 reflect poor prognosis and numbers less than one reflect the opposite."

We thank the editor for their feedback. We have attended to this statistical analysis and reporting as suggested.

"The Figures 2-6 must include the outcome being measured."

We thank the editor for their comment, we have amended the figure titles to reflect this and note that these figures 2-6 will be presented alongside the figure titles/legends to avoid any confusion about the outcome.

"Table 3. Including the Effect size without 95% CI is meaningless for a reader. It should be RR with 95% CI demonstrating that none are close to statistically significant."

We thank the editor for their comments. We have updated table 3 as suggested to incorporate these parameters.

Reviewer comments

"In the introduction part, I'd emphasize that Atlanta classification is not used to predict the severity of AP, but it's a classification system with clearly defined criteria for mild, moderate, and severe AP."

We thank the reviewer for their comment. We agree with this and have therefore amended the introduction to reflect this.

"There is very little data on the impact of sarcopenia on AP outcomes, and it is not clear whether sarcopenia is a cause or a consequence in patients with severe pancreatitis. The fact is that more than 25% of patients with AP have sarcopenia on admission. However, it would be very important to determine to what extent sarcopenia changes during hospitalisation and how the possible worsening of sarcopenia affects the outcome in AP."

We thank the reviewer for their feedback and for raising this important and clinically relevant research question. The fact of the matter is that of the included studies, only 1 explored the temporal impact of body composition changes on acute pancreatitis, and this was in the necrotising pancreatitis cohort. Essentially, the purpose of this systematic review and the remaining studies were to identify sarcopenia at start of admission as a prognostic indicator of outcomes. Therefore, the question about temporal changes in sarcopenia/degree of sarcopenia, particularly with severe cases of acute pancreatitis remains underexplored. We have incorporated comments related to this in the discussion and highlight this as a gap in the literature.

"CT is a fairly reliable method for detecting sarcopenia, but it is very important to point out that CT in AP is very often not performed on admission of the patient, but only after a few days. The aforementioned period in severe AP can only worsen the sarcopenia compared to the period of the patient's admission to hospital or the onset of symptoms."

We thank the reviewer for this comment which aligns very closely with the aforementioned comment on sarcopenia and body composition change during admission. We have attended to this by considering the impact of this variable into the discussion.

"There is also the question of the appropriateness of nutritional intervention in these patients."

We thank the reviewer for their comment. Nutritional interventions remains an area of ongoing research however it is an important consideration. As suggested, comments related to this have been incorporated into the discussion.

"Most patients with AP, especially in retrospective analyses, actually have a mild to moderate form of the disease, and even when the authors state that the predicted severe forms of the disease are included in the study, a large proportion of these patients do not ultimately have severe AP. In these patients, sarcopenia is unlikely to have a significant impact on the outcome of the disease."

We thank the reviewer for their comment. We agree that this is the case and therefore have included comments about the study population and the severity of disease, particularly in relation to subgroup analyses and our inability to do this due to the limitations in data.

"It would also be good to include the presence of comorbidities in the data analysis, because other diseases, especially chronic diseases, can significantly influence the presence of sarcopenia and the outcome of AP. In addition to the limitations of your research already mentioned, I think the discussion should be expanded to include the above comments."

We thank the reviewer for their comment. We agree the presence of comorbidities can also influence the development of sarcopenia. We note however that these comorbidities were underreported in the literature and therefore have raised this as a limitation of this underlying evidence.

---

## [Editor Report · Decision Letter 1]

21 Mar 2025

A systematic review on the prognostic role of radiologically-proven sarcopenia on clinical outcomes of patients with acute pancreatitis.

PONE-D-24-32431R1

Dear Dr. Le,

We’re pleased to inform you that your manuscript has been judged scientifically suitable for publication and will be formally accepted for publication once it meets all outstanding technical requirements.

Kind regards,

James M Wright

Academic Editor

PLOS ONE
---

## [Editor Report · Acceptance letter]

PONE-D-24-32431R1

PLOS ONE

Dear Dr. Le,

I'm pleased to inform you that your manuscript has been deemed suitable for publication in PLOS ONE. Congratulations! Your manuscript is now being handed over to our production team.

Kind regards,

on behalf of

Professor James M Wright

Academic Editor

PLOS ONE